# Enabling On-Device Large Language Models with 3D-Stacked Memory

**Lita Yang**    **Kavya Sreedhar**[*]    **Huichu Liu**    **Edith Beigné**
Reality Labs, Meta, Inc.
Sunnyvale, CA, USA
yanglita@meta.com

## Abstract

In this paper, we address the growing need for new types of memories to enable deployment of on-device large language models (LLMs) to resource-constrained augmented reality (AR) edge devices. We evaluate the memory power and area savings using 3D-stacked memory (3D-DRAM, 3D-SRAM) versus conventional 2D memory (LPDDR-DRAM, SRAM). At target inference rates of 5-100 inferences per second, 3D-DRAM consumes the least memory power across all the memory options, achieving ∼7-15x improvement in memory power consumption compared with conventional 2D memory across our benchmark suite of on-device LLMs (Distilled GPT-2, GPT-2, BART Base, and BART Large). While 3D-SRAM can reduce memory dynamic power, the leakage power consumption for storing such a large model becomes prohibitively costly, hence why 3D-DRAM becomes a better option than 3D-SRAM for on-device LLMs. Additionally, since 3D-DRAM significantly reduces the memory power consumption for on-device LLMs to 10's of mWs, 3D-DRAM enables the deployment of much larger LLMs that previously could not be deployed with conventional DRAM and 2D SRAM solutions.

## 1   Introduction and Motivation

Modern augmented reality (AR) and edge devices are integrating more and more AI/ML capabilities. With recent advancements in large language models (LLMs), the feasibility of using one multimodal AI model on AR devices to enable a smart and context-aware AI assistant is becoming more of a reality [1, 2, 3]. AR wearable devices, however, are highly resource constrained and require major technological innovations to meet the strict real-time latency, power, and area requirements while enabling key user experiences [4] such as multimodal AI. Integrating LLMs on-device is not a easy task, as even a LLM such as LLaMA 7B [5] with 8-bit weights can exceed the low single GB's of DRAM allocated for AR devices and wearables. Additionally, factoring in LLM energy consumption (∼0.1 J/token per billion in model parameters [6, 7]), a 7B LLM consumes ∼0.7 J/token, which greatly exceeds power budget requirements for battery-powered edge devices [7].

Notably, LLMs tend to be highly memory-bound and improving memory bandwidth and reducing memory power consumption of LLM inference [8, 9] is a key enabler of on-device LLM deployment. Figure 1 illustrates the memory hierarchy of conventional edge devices. Currently, for on-device LLM use cases which are expected to fit in the form factor of <200 MB on AR glasses and consume <100 mW [10], there exists a gap between low capacity on-chip SRAM and power hungry off-chip LPDDR memory to meet the power budget and capacity needs of deploying LLMs for AR devices. Off-chip LPDDR-DRAM end-to-end memory power is significantly high (∼85 pJ/B) and pushes current edge devices to include large on-chip SRAMs to reduce the number of off-chip memory accesses. However, scaling on-chip SRAMs to mitigate off-chip DRAM power and latency shortcomings is increasingly

---

[*]Work done while author was an intern at Meta

38th Second Workshop on Machine Learning with New Compute Paradigms at NeurIPS 2024 (MLNCP 2024).

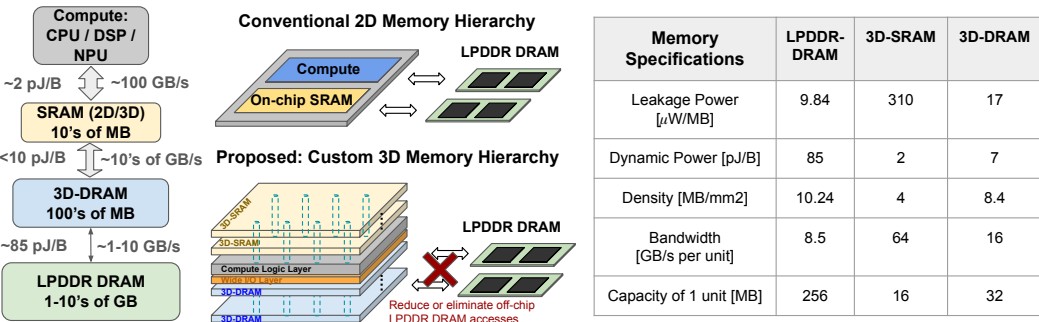

| Memory Specifications | LPDDR-DRAM | 3D-SRAM | 3D-DRAM |
|---|---|---|---|
| Leakage Power [μW/MB] | 9.84 | 310 | 17 |
| Dynamic Power [pJ/B] | 85 | 2 | 7 |
| Density [MB/mm2] | 10.24 | 4 | 8.4 |
| Bandwidth [GB/s per unit] | 8.5 | 64 | 16 |
| Capacity of 1 unit [MB] | 256 | 16 | 32 |

Figure 1: Memory hierarchy for conventional 2D edge devices versus our proposed 3D-stacked memory hierarchy. The table provides our memory modeling specifications for the three types of memories being considered (Conventional: LPDDR-DRAM, versus 3D-Stacked Memories: 3D-DRAM and 3D-SRAM).

a costly solution, as on-chip SRAM can consume a significant portion of die area, SRAM area is not scaling with process nodes, and leakage power can become significant for large SRAM capacities. Additionally, many on-device AI applications need to be always-on so techniques like power gating to reduce SRAM leakage may not necessarily help or be applicable. Ideally, we would like to have a memory solution with power and bandwidth close to SRAM, leakage and density close to DRAM, and options for a more scalable physical footprint.

Because of this, a new class of ultra-low power and high bandwidth memory optimized for on-device AI is necessary to enable the deployment of LLMs for wearable devices, especially for AR. As shown in Figure 1, we propose using 3D-stacked memory to integrate one or multiple memory dies on top of the logic die in the vertical dimension, allowing for high bandwidth and ultra-low power 3D connections while achieving the same or smaller footprints with larger memory capacities [10]. With the goal of enabling on-device LLMs on AR devices for privacy and real-time latency considerations, we quantify the benefits of using 3D-stacked memory compared to conventional 2D DRAM and SRAM solutions and analyze the trade-offs between different memory hierarchies with 3D-stacked memories. Since LLMs and Transformer-style models are typically memory bound [8, 9], our analysis focuses on memory power and area for LLM inference, which can easily become the dominant consumer of total AR device power and area.

In this paper, we demonstrate benefits of adding 3D-DRAM for lower memory power for on-device LLM use cases, opportunities to reduce and alleviate large SRAM area on-chip, and reduce/eliminate expensive off-chip accesses to LPDDR-DRAM. Overall, 3D-DRAM can provide ∼7-15x improvement in memory power consumption over conventional 2D memory for on-device LLMs <200M parameters and more notably, reduces memory power to acceptable ranges of 10's of mW for AR devices. Additionally, the reduction in memory power allows us to deploy larger variants of LLMs (BART Large vs. BART Base, GPT-2 vs. Distilled GPT-2) not previously feasible in the constraints of AR power budgets with conventional 2D memory. From an area perspective, 3D-DRAM is also more competitive than conventional 2D memory options since 3D-stacking enables continued scaling of on-device memory capacity in the vertical dimension.

## 2 Methods and Evaluation Setup

### 2.1 Models and Use Cases

To analyze on-device LLM use cases which can reasonably fit in the form factor of <200 MB, we model four on-device LLMs targeting deployment on AR devices as shown in Table 1: Distilled GPT-2 [11], GPT-2 [12], BART Base [13], and BART Large [13]. While there are newer variants of LLMs such as MobileLLM [7] and MiniLLM [14] which target on-device mobile use cases <1B parameters, these models are still too large and power hungry for the stringent form factors and budgets of AR devices. We target LLMs <200M parameters (∼200 MB with quantization to 8-bits) to analyze the feasibility of edge deployment given AR device footprint and power limitations. Note that while <200M parameter LLMs may not be as accurate as their larger variants in mobile or

Table 1: On-device LLMs Evaluation Benchmark Suite <200M parameters

| Models | Distilled GPT-2 [11] | GPT-2 [12] | BART Base [13] | BART Large [13] |
|---|---|---|---|---|
| Model Footprint (8-bits) | 79 MB | 119 MB | 88 MB | 195 MB |
| GFLOPs | 1.37 | 2.74 | 1.67 | 4.85 |
| Operational Intensity (Ops/B) | 16.1 | 20.9 | 17.6 | 22.9 |

cloud, we see this becoming more feasible as new distillation and compression techniques are getting better [7, 14]. For sequence length, we constrain to short sequence lengths of 16 since on-device use cases generally involve short message responses and quick summarizations [2, 3]. We leave to future work to analyze use cases in which much longer sequence lengths are necessary.

Table 1 summarizes the memory capacities needed for these models and illustrates that they are generally memory bound, since the operational intensity or number of operations per byte (Ops/B) is small (<25 Ops/B). Given these models are memory-bound, we focus this work on optimizing the memory aspects of on-device LLM deployment by: (1) analyze/quantify the memory power reduction achievable using 3D-stacked memories (3D-DRAM, 3D-SRAM) and (2) demonstrate the feasibility of deploying larger LLMs not previously possible in the stringent power budget and footprint constraints of AR devices.

## 2.2   3D-Stacked Memory Modeling Parameters

We investigate two types of advanced 3D-stacked memory, 3D-SRAM and 3D-DRAM, as shown in Figure 1, compared with conventional LPDDR-DRAM and/or SRAM solutions (our 2D baselines). We use the memory modeling specifications in the table of Figure 1 for the three different memory options, 3D-DRAM, 3D-SRAM, and LPDDR-DRAM, to perform our analysis. We assume LPDDR-DRAM is based off of LPDDR4X technology, 3D-SRAM numbers in 7nm technology were obtained from [15], and 3D-DRAM numbers [10] use specifications based off DRAM technology but optimized for much lower dynamic power due to a custom wide-direct, PHY-less, low pin-speed interface and controller. Note that 2D SRAM power numbers are similar to 3D-SRAM as shown in [15, 16] but thanks to 3D-stacking, 3D-SRAM can have much smaller footprints. From Figure 1, we see that conventional LPDDR-DRAM has high cell density and low leakage power but consumes high dynamic energy. 3D-SRAM has the lowest dynamic energy but has high leakage power and the lowest density. 3D-DRAM is a trade-off between the two other memory technologies, but consumes <10 pJ/B memory power access (∼12x lower than LPDDR-DRAM) with a balance of slightly higher leakage power while achieving similar memory density to LPDDR-DRAM.

We explored one-level and two-level memory hierarchies with the 3D-stacked memory options. To determine how to utilize the two levels in the memory hierarchy, we evaluate: (1) storing parameters and activations for layers with lower Ops/B in the larger memory and (2) storing all parameters in the larger memory and activations in the smaller memory. We found that strategy (1) yielded two-level memory hierarchies with prohibitively large SRAM sizes (e.g., 48 MB SRAM for Distilled GPT-2 and 80 MB SRAM for GPT-2) that are not as reasonable from an AR device form-factor perspective, while strategy (2) resulted in more reasonable SRAM sizes (16 MB for the GPT-2 models). Thus, our evaluation going forward will utilize strategy (2) for determining where to store data in the two-level memory hierarchies evaluated. We benchmark against two baselines:

- **LPDDR-DRAM:** All data stored in LPDDR4X DRAM
- **LPDDR-DRAM + $X$ MB SRAM:** Two-level 2D memory hierarchy with LPDDR4X DRAM and $X$ MB of SRAM

Then we compare against three 3D-stacked memory options:

- $X$ **MB 3D-SRAM:** All data is stored in $X$ MB of 3D-SRAM
- $X$ **MB 3D-DRAM + $Y$ MB 3D-SRAM:** Two-level memory hierarchy with $X$ MB of 3D-DRAM and $Y$ MB of 3D-SRAM
- $X$ **MB 3D-DRAM:** All data stored in $X$ MB of 3D-DRAM

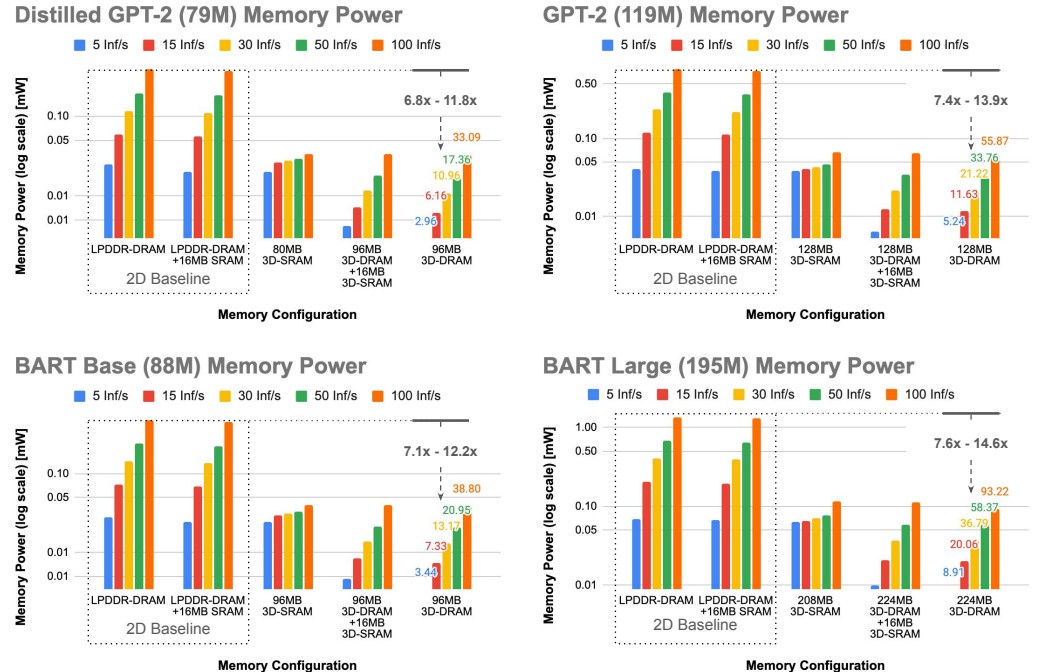

Figure 2: Memory power consumption for on-device LLMs across target inference rates of 5 - 100 Inf/s. The lowest memory power point is highlighted and consistently shows that 3D-DRAM provides the optimal memory power consumption for these models.

The values of $X$ and $Y$ are set based on the minimum memory size required to support the given model footprint, given the unit capacities assumed from the table of Figure 1.

## 2.3 Modeling Tool

An in-house modeling tool was built in Python to import pre-trained PyTorch models, extract the layers, calculate the dimensions per layer, memory requirements, FLOP count, and operations per byte. We assume all parameters and activations can be quantized to 8-bits, and the model parameters include both the weights and biases. We assume a weight-stationary dataflow for the architecture, in which we calculate the memory required for the model parameters and only the activations for the layer with the largest activation size. Since our goal is to estimate the benefits of using 3D-stacked memory versus conventional 2D memory, this modeling assumption provides sufficient high-level estimation for our purposes.

## 3 Results and Analysis

In this section, we sweep and analyze our modeling results for our on-device LLM benchmark suite. Since multimodal on-device AI use cases and requirements can vary widely and are constantly being redefined, we consider a broad range of target inference rates from 5-100 inferences per second (Inf/s) to understand the scenarios in which 3D-stacked memory is most beneficial.

**Memory Power Savings Using 3D-DRAM** Figure 2 summarizes the total memory power consumption for our on-device LLM benchmark suite across the target range of inference rates. Compared to LPDDR-DRAM and the hybrid LPDDR-DRAM + 16MB of SRAM memory configurations (2D memory baselines), 3D-DRAM consumes the lowest memory power for all target inference rates, achieving 6.8 - 14.6x improvement in memory power consumption compared with the conventional 2D baselines. For higher target inference rates (>30 Inf/s), the memory power consumption of the 3D hybrid option (3D-DRAM + 16MB of 3D-SRAM) comes close to the memory power consumption of the 3D-DRAM only option, indicating that for higher inference rates, some on-chip SRAM may

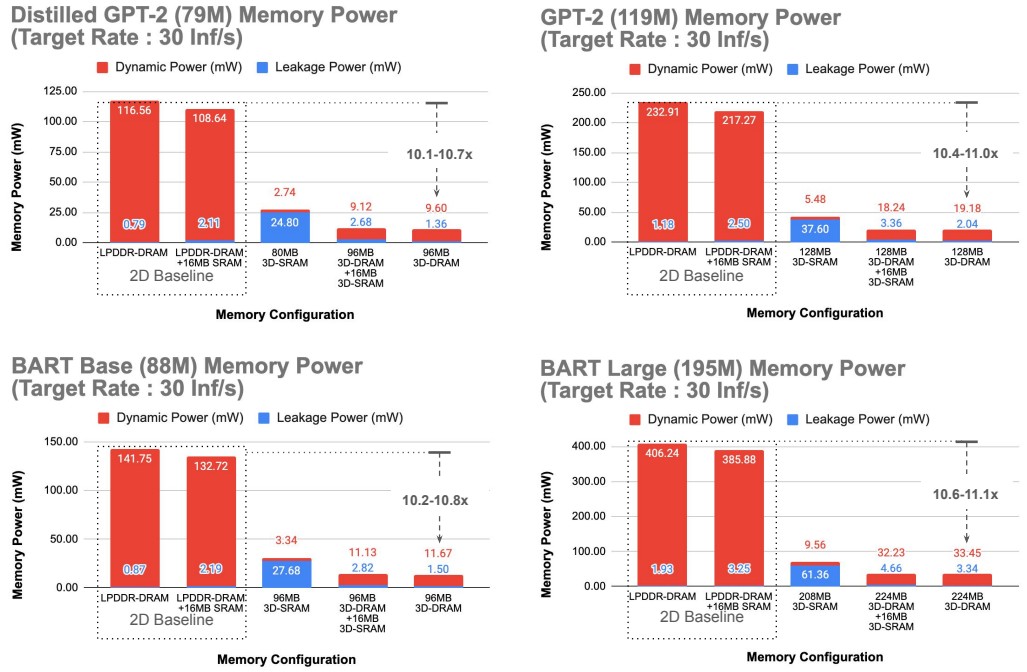

Figure 3: Memory power breakdown for on-device LLMs for target inference rate of 30 Inf/s. SRAM leakage power becomes dominant as you scale up in memory capacity, leading to diminishing returns on increasing SRAM sizes for optimal memory power consumption.

be beneficial for speed considerations. However, for these target rates and memory capacities, 3D-DRAM is the clear winner in terms of lowest memory power consumption compared to all of the other memory configurations.

Additionally, we see that not only does 3D-DRAM achieve reductions in memory power consumption across the suite of on-device LLMs compared to conventional 2D memory baselines, it significantly reduces memory power consumption to 10's of mW. This is critical for battery-powered AR devices in which <100 mW of power consumption would be ideal but is often challenging for deploying LLMs on-device. The 2D baseline memory hierarchy options significantly exceed the memory power budget for GPT-2 and BART Large (>100 mW), but 3D-DRAM reduces the memory power consumption to 5 - 93 mW. Enabling deployment of these larger models on edge devices allows for improved model accuracy compared to the smaller counterparts for these models (i.e., Distilled GPT-2 and BART Base).

**3D-DRAM vs. 3D-SRAM Trade-off**   To understand why the 3D-SRAM only memory configuration and the hybrid 3D-DRAM and 3D-SRAM solution is not as competitive with the 3D-DRAM only solution, Figure 3 dives deeper into one of the target inference rates, 30 Inf/s, which is in the middle of the target ranges. For a target inference rate of 30 Inf/s, 3D-DRAM only consumes the lowest power across these workloads with a range of model footprint sizes, achieving ∼10-11x lower power compared to the 2D memory baselines. When observing the memory power breakdown between dynamic and leakage power, we note that while 3D-SRAM reduces memory dynamic power significantly, the leakage power consumption for storing such a large model becomes dominant, hence why 3D-DRAM becomes a better option than 3D-SRAM at these memory capacities.

When comparing the hybrid 3D-DRAM + 16MB 3D-SRAM option with 3D-DRAM only, we note dynamic power is competitive but the hybrid option with 3D-SRAM adds additional leakage, making the hybrid option less attractive. Additionally, in the case of BART Large, 224 MB of 3D-DRAM is required due to the 3D-DRAM unit capacity of 32 MB, while only 208 MB of SRAM is required due to a SRAM unit capacity of 16 MB (taken from the table of Figure 1). However, this slightly larger 224 MB 3D-DRAM is still better from a power and area perspective compared to the smaller 208

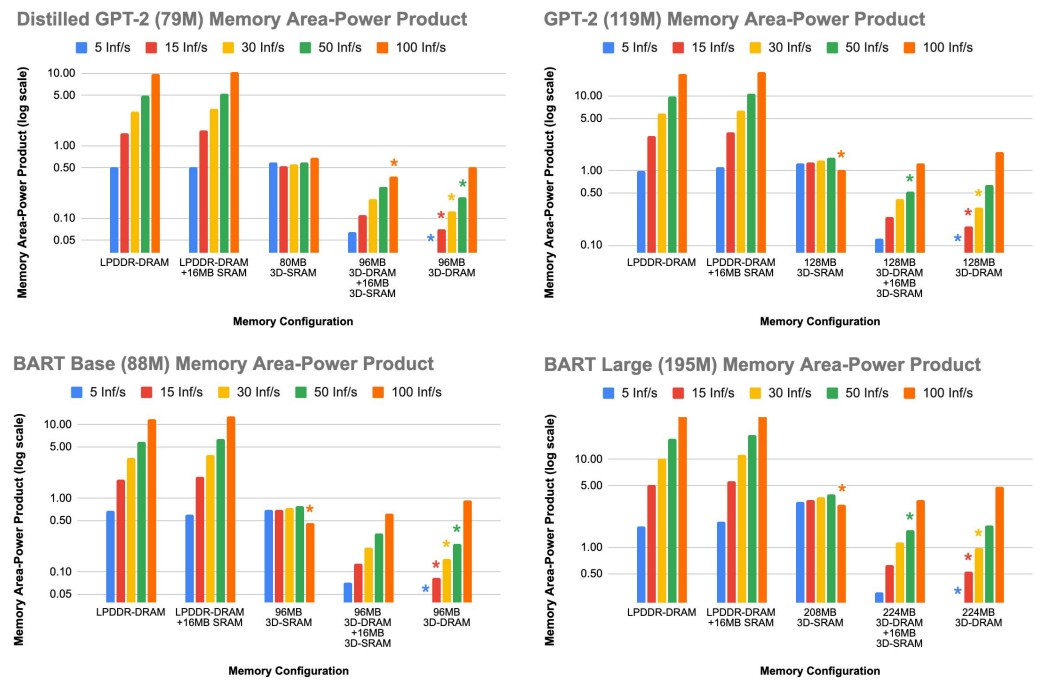

Figure 4: Memory area-power product figure of merit across the benchmark suite of on-device LLMs for the target inference rates of 5 - 100 Inf/s. "*" indicates the optimal configuration point.

MB 3D-SRAM since 3D-DRAM requires ~18x lower leakage power, and 3D-DRAM is far more dense than 3D-SRAM (~2x).

**Case Study: using Figure of Merit = Area-Power Product**   Since AR devices are very area-limited, we propose using a figure of merit weighting memory area and power equally (area x power) similar to [16] to find the sweet spot for optimizing both memory power and area. We plot in Figure 4 the area-power product across the range of target inference rates and workloads to understand the point at which 3D-SRAM and/or hybrid 3D-DRAM + 3D-SRAM configurations may become competitive with 3D-DRAM only. Figure 4 highlights the lowest memory area-power products across the suite of workloads and target inference rates. At 50-100 Inf/s, we start to see the 3D-SRAM only and 3D-DRAM + 3D-SRAM hybrid options become more competitive from both a memory power and area optimization objective, while lower inference rate (<50 Inf/s) still favor the 3D-DRAM only memory configuration.

## 4   Conclusion

In this paper, we present benefits of using 3D-stacked memory for reducing memory power consumption for on-device LLMs. At target inference rates of 5-100 inferences per second, 3D-DRAM consumes the lowest memory power across all the memory options, achieving ~7-15x improvement in memory power consumption compared with the conventional 2D memory across our benchmark suite of on-device LLMs (Distilled GPT-2, GPT-2, BART Base, and BART Large). While 3D-SRAM can reduce memory dynamic power, the leakage power consumption for storing such a large model becomes dominant, hence why 3D-DRAM becomes a better option than 3D-SRAM for on-device LLMs. If inference speed becomes critical for these applications, however, we note that from an area-power perspective, it may be optimal to use 3D-SRAM + 3D-DRAM hybrid memory hierarchies. Finally, since 3D-DRAM significantly reduces the memory power consumption for on-device LLMs to 10's of mWs, 3D-DRAM enables the deployment of much larger LLMs that previously could not be deployed with conventional DRAM and 2D SRAM solutions.

## Acknowledgments

The authors would like to thank Daniel Morris who helped provide feedback on 3D-DRAM technology and discussions on the modeling assumptions and memory power results. We would also like to acknowledge Juhyoung Lee (former intern) who started the initial analysis on 3D-DRAM for Transformer/Assistant AI workloads, which we later expanded to on-device LLMs.

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
