# OpenReview forum: "Enabling On-Device Large Language Models with 3D-Stacked Memory"
_NeurIPS.cc/2024/Workshop/MLNCP — MLNCP Poster_

### Official Review · Reviewer_fbwQ · 2024-09-26
**Provides novel solution to existing problems**

**Rating:** 7
**Confidence:** 3

**Review:**

The paper explores the use of 3D-stacked memory (DRAM and SRAM) to enable efficient deployment of large language models (LLMs) on augmented reality (AR) and edge devices. The paper proposes attempts to combat the large die area requirement of SRAMs and high-power consumption of off-chip LPDDR-DRAM.

Through the experiments the authors successfully showcase the significantly lower memory power consumption for 3D-stacked memory setups. They showcase the power consumption over a variety of target inference rates which provides an understanding of which 3D setup provides the best performance for different compute requirements.

Some improvements/questions that the authors can address in the future include :

Include Performance Metrics: The analysis focuses primarily on power savings given some target inference rate, but how does the use of 3D-SRAM and 3D-DRAM affect the inference latency and throughput of the large language models? Can you provide more details on how these memory configurations influence overall performance in terms of response times?

Scalability to Larger Models: Although the paper effectively demonstrates power savings for models under 200 million parameters, however, as the trend is moving towards LLMs with billions of parameters, how scalable is the 3D-stacked memory solution will be for much larger models?

Real-World Deployment Considerations: What are the foreseeable practical challenges in deploying 3D-DRAM or hybrid memory configurations in commercially available AR or wearable devices?

---

### Decision · Program_Chairs · 2024-10-10

Accept (Poster)